# Syntrophic Hydrocarbon Degradation in a Decommissioned Off-Shore Subsea Oil Storage Structure

**DOI:** 10.3390/microorganisms9020356

**Published:** 2021-02-11

**Authors:** Adrien Vigneron, Perrine Cruaud, Frederic Ducellier, Ian M. Head, Nicolas Tsesmetzis

**Affiliations:** 1Shell International Exploration and Production Inc., Houston, TX 77082, USA; 2School of Civil Engineering and Geosciences, Newcastle University, Newcastle upon Tyne NE1 7RU, UK; ian.head@newcastle.ac.uk; 3Institut de Biologie Intégrative et des Systèmes, Université Laval, Québec, QC G1S 0A6, Canada; perrine.cruaud@gmail.com; 4Société des Pétroles Shell, 92708 Colombes, France; Frederic.Ducellier@shell.com

**Keywords:** MAGs, metagenomics, oil systems, bioremediation, marine environments, alkanes, microorganisms

## Abstract

Over the last decade, metagenomic studies have revealed the impact of oil production on the microbial ecology of petroleum reservoirs. However, despite their fundamental roles in bioremediation of hydrocarbons, biocorrosion, biofouling and hydrogen sulfide production, oil field and oil production infrastructure microbiomes are poorly explored. Understanding of microbial activities within oil production facilities is therefore crucial for environmental risk mitigation, most notably during decommissioning. The analysis of the planktonic microbial community from the aqueous phase of a subsea oil-storage structure was conducted. This concrete structure was part of the production platform of the Brent oil field (North Sea), which is currently undergoing decommissioning. Quantification and sequencing of microbial 16S rRNA genes, metagenomic analysis and reconstruction of metagenome assembled genomes (MAGs) revealed a unique microbiome, strongly dominated by organisms related to *Dethiosulfatibacter* and *Cloacimonadetes*. Consistent with the hydrocarbon content in the aqueous phase of the structure, a strong potential for degradation of low molecular weight aromatic hydrocarbons was apparent in the microbial community. These degradation pathways were associated with taxonomically diverse microorganisms, including the predominant *Dethiosulfatibacter* and *Cloacimonadetes* lineages, expanding the list of potential hydrocarbon degraders. Genes associated with direct and indirect interspecies exchanges (multiheme type-C cytochromes, hydrogenases and formate/acetate metabolism) were widespread in the community, suggesting potential syntrophic hydrocarbon degradation processes in the system. Our results illustrate the importance of genomic data for informing decommissioning strategies in marine environments and reveal that hydrocarbon-degrading community composition and metabolisms in man-made marine structures might differ markedly from natural hydrocarbon-rich marine environments.

## 1. Introduction

Hydrocarbons are among the most abundant small organic molecules on Earth, particularly in anoxic marine sediments and deep subsurface environments [1]. Their presence is a strong structuring factor for the prokaryotic and eukaryotic communities, providing an abundant carbon source for specific hydrocarbon-degrading organisms while inhibiting others due to their toxicity [1]. In open seawater, hydrocarbons are mainly degraded by aerobic lineages such as some members of the Gammaproteobacterial orders *Oceanospirillales* and *Alteromonadales* through the production of alkane mono-oxygenase, encoded by *alkB* genes [2,3]. However, in constrained marine environments, such as sediments and oil seeps, oxygen is rapidly depleted, and hydrocarbons are mainly degraded by anaerobic metabolisms [4]. Most of the anaerobic hydrocarbon-degrading lineages cultivated and known to date belong to the *Deltaproteobacteria* (e.g., *Syntrophus*, *Smithella*, *Desulfoglabea* and *Desulfobacteraceae*) and *Clostridiales* classes [4,5]. Genomic analyses of these cultivated species have led to the identification of key genes and enzymes involved in anaerobic hydrocarbon activation and degradation [6]. Although there are several pathways for anaerobic hydrocarbon degradation, enzymes of the Alkyl-succinate synthase family, encoded by *assA*, *bssA* or *nmsA* genes, are widespread among anaerobic hydrocarbon degrading bacteria and encode the alpha subunit of enzymes required for the initial activation of alkanes, toluene and naphthalene, respectively [4]. Amplification and sequencing of alkyl-succinate synthase genes from marine sediments demonstrated a large pool of hydrocarbon degradation gene variants, suggesting that unidentified anaerobic hydrocarbon degrading species must exist [7]. Potential hydrocarbon degrading lineages are also frequently identified in metagenomic datasets from marine and oil-rich environments, suggesting that the inventory of anaerobic hydrocarbon-degrading microbial taxa is far from complete [8,9,10]. However, our understanding of the exact nature of anaerobic hydrocarbon degradation processes is challenged by the complex interspecies interactions that are frequently observed in hydrocarbon degrading enrichments [11]. Syntrophic metabolisms and interspecies transfer of energy through the exchange of metabolic shuttles such as acetate, formate or hydrogen are common in anaerobic hydrocarbon degrading enrichments, leading to complex microbial networks between taxonomically diverse microorganisms [4,12]. In addition, syntrophic metabolism might also involve direct interspecies electron transfer through conductive nanowires such as in the aromatic hydrocarbon degrader genus *Geobacter* [13] and multiheme cytochromes as observed in some anaerobic methane oxidizing consortia [14].

Extraction of oil from reservoirs has been found to induce major changes in the oil field microbiome, by introducing exogenous electron acceptors and microorganisms into the subsurface reservoir [9,15]. These human-driven perturbations have major consequences for the endemic oil reservoir microbiome, modifying community composition and function, with important aftereffects such as elevated production of H_2_S, representing an increased health and safety hazard for topside personnel, reduction in oil quality and value through oil souring and asset’s structure integrity due to microbiological influenced corrosion [16]. These impact oil production and increase the risk of environmental pollution. Nevertheless, due to practical difficulties, mostly related to sampling, microbiological analyses of offshore facilities are infrequent and primarily focused on microbial control for corrosion prevention, leaving the microbiome from other parts of the infrastructure underexplored.

Gravity-based substructures (GBS) are steel reinforced concrete constructions that stand on the seabed and are held there by gravity alone, providing support for offshore structures such as oil platforms and wind turbines at relatively shallow water depths (average water depth of 113 m). These structures typically contain storage tanks (often termed “cells”) for temporary storage of oil before it is transferred to shore for processing. The Brent field, an oil and gas field located in the North Sea, has been operated by four platforms, three of which are gravity-based structures. Following over 40 years of production, the Brent field has now recovered over 99.5% of its economically recoverable reserves. It has reached the end of its life and the decision to now decommission Brent is a natural step in the life cycle of the field. As part of the decommissioning process, it was necessary for operators to evaluate the best strategy for safe decontamination of residual hydrocarbons associated with the oil field infrastructure. On the basis of many years of detailed study and assessment, it has been concluded that there are no practically available options for removing the whole of a Brent GBS, either in one piece or several pieces. The owners consider that a safer alternative is to remove all the attic oil from within the storage cells and then leave the structure to degrade naturally over time. This approach raises the question of whether the cell contents would pose a risk when they will eventually be released into contact with the marine environment.

Combining metagenomic and chemical analyses, we investigated the microbiome of these man-made submarine structures. We aimed to evaluate the hydrocarbon biodegradation potential of the microbial community to assess the potential for natural attenuation of hydrocarbon contamination in this ecosystem that could remain isolated for up to 250 years (minimal lifetime of concrete substructures) once the decommissioning process, which will leave the structures in place, is completed. In the light of our results, we assessed the benefits of using metagenomic approaches to evaluate the biodegradation potential of oil system microbiomes.

## 2. Materials and Methods

### 2.1. Study Site and Sampling

The Brent field is an oil and gas field located in the North Sea, 186 miles northeast of the Shetland Islands at water depth of 140 m. Four platforms, Alpha, Bravo, Charlie and Delta, have facilitated the production operations in the Brent field during the 40 years of its production life. Delta platform has a total of 16 storage cells that allowed oil and water phases of the production fluids to separate. The water column of central concrete gravity base structure (Cell6, 16,470 m^3^, Figure 1) was sampled in June 2016 using three pressurized containers (600 cm^3^ each). Containers were submerged to approximately halfway the aqueous phase of Cell6 (i.e., approximately 20 m from the top), and then the inlet of the sampling container was opened, allowing fluids to enter. Upon filling up, the inlet was closed, and the container was raised to the surface. Three samples (Cell6.2, Cell6.3 and Cell6.4) were taken within a few hours of each other and placed inside cool boxes with ice packs until taken on shore. Once on shore, the containers were stored at 5 °C for two weeks before processing. Cell6.2 was used for water chemical analysis performed in duplicates by SGS laboratories (Billingham, UK). Petroleum hydrocarbon concentrations were analyzed by Gas Chromatography with Flame-Ionization Detection (GC-FID) using TNRCC method 1006. Benzene, ethyl-benzene, toluene, xylene and volatile fatty acids (acetic, propanoic, valeric and hexanoic acids) concentrations were determined by Gas Chromatography coupled to Mass Spectrometry (GC-MS) using the US-EPA-624 method. Naphthalene concentration was also determined using GC-MS and the US-EPA-625 method. Total iron and manganese concentrations were measured by Inductively Coupled Plasma Mass Spectrometry (ICP-MS) following US EPA 200.7 method, while nitrate, nitrite and sulfate concentrations were measured by ion chromatography according to the ISO 10304 protocol. Presence of hydrogen sulfide in the water was estimated by lead-acetate test paper according to ASTM D6021-06 instructions (Appendix A).

Contents of the remaining two pressurized containers (Cell6.3 and Cell6.4) were filtered on 0.22 µm Sterivex filters (EMD Millipore, Darmstadt, Germany). Fluids from each pressurized container were filtered in triplicates (60 mL each). Nucleic acids on the filters were preserved by the addition of 5 mL of RNAlater (Sigma-Aldrich, Saint Louis, MO, USA). DNA was extracted from two of the three replicates (i.e., Cell6.3a, Cell6.3b, Cell6.4a and Cell6.4b) using the FastDNA Spin kit for Soil (MP Biomedicals, Santa Anna, CA, USA), according to the manufacturer’s recommendations, leading to an average of 2.19 ng of DNA per µL. The remaining replicate remained stored as backup sample. The extracted DNA was stored at −20 °C until further analysis.

### 2.2. Illumina MiSeq Library Preparation, Sequencing and Analysis

Microbial community composition of all samples was determined by high throughput sequencing of bacterial and archaeal 16S rRNA genes using prokaryotic primers modified from Parada et al. [17]: U515New_F: 5′-GTG YCA GCM GCC GCG GTA A-3′ and U926New_R: 5′-CCG YCW ATT YMT TTR AGT TT-3′. The primer sequence was modified to enhance the coverage of the oligonucleotides over *Epsilonproteobacteria*/*Campylobacteria* that are frequently detected in oil systems [9]. In silico analysis of the primers is available in the Appendix A). All PCR reactions were conducted in triplicate with negative controls as follows: 35 cycles of denaturation at 95 °C for 30 s and then 30 s of annealing at 55 °C and 30 s of extension at 72 °C. Afterwards, amplicons from the triplicate reactions were pooled and purified from agarose gels using Qiagen’s MinElute Gel purification kit (Qiagen, Hilden, Germany). PCR products were indexed using a Nextera XT kit (Illumina Inc., San Diego, CA, USA) according to the manufacturer’s recommendations and sequenced on an Illumina MiSeq sequencer (Illumina Inc., San Diego, CA, USA), using an Illumina MiSeq v3 kit. Sequences were filtered for quality, pair-end joined and clustered into operational taxonomic units (OTUs) as previously described (10). Then, taxonomic affiliations of the reads were determined using Mothur’s Bayesian classifier [18] with the Silva database release 132 as reference [19]. The results were rarefied using the sub.sample command of the Mothur pipeline to have the same number of sequences per sample (15,515 sequences, corresponding to the lowest number of sequences in one sample).

Metagenomes for samples Cell6.3a and Cell6.4a were also constructed using the Nextera XT Library Kit (Illumina, San Diego, CA, USA) according to manufacturer’s recommendations. Tagmentation and indexing were checked using a High Sensitivity DNA chip on an Agilent Bioanalyzer 2100 (Agilent Technologies, Santa Clara, CA, USA). Metagenomes were normalized and diluted to 4nM based on the average size and concentration obtained from the Bioanalyzer. The two metagenomes were pooled in equimolar quantities prior to sequencing. Metagenome libraries were sequenced using an Illumina MiSeq V3 kit (Illumina, San Diego, CA, USA). Barcodes and adapters from the produced metagenomic datasets were trimmed on-instrument using Illumina’s MiSeq Reporter software (Illumina Inc., San Diego, CA, USA) and downloaded as fastq files. Datasets were quality filtered using Trimmomatic [20], using default setting for paired-end Illumina data. Paired-end joining was done using the “join_paired_ends.py” script bundled with the QIIME 1.9.1, using default settings [21], leading to an average of 22,357,841 quality-filtered and paired-end sequences per samples. Assembly was performed from paired-end joined reads passing quality filtering using IDBA-UD [22], using default settings. After assembly, all reads which passed quality filtering were mapped back to the assembled contigs using BBMap [23] to determine the average sequencing depth of each contig. Reads longer than 200 bp that were not mapped to the assembly were combined with the assembled contigs into a single fasta file before upload to the IMG/M analysis pipeline for gene calling and functional annotation [24]. Metagenomes were normalized prior to sample comparison using MUSiCC, with inter-sample variation mode [25]. For a primer-free analysis of the microbial community composition, 16S rRNA genes were extracted from the metagenomic datasets using REAGO 1.1 [26], leading to an average of 34,710 16S rRNA reads per sample, and then taxonomic affiliation of the recovered sequences was carried out using blastn against the Silva 132 database. 16S rRNA genes were assembled from sorted 16S rRNA reads using SPADES (careful mode) [27], allowing sequence comparisons over longer (>800 bp) to full length 16S rRNA gene. Metagenomes are available in IMG/M under the following accession numbers: 3300010410 and 3300010411. For genomic binning, all quality filtered sequences were pooled and co-assembled using MEGAHIT [28]. Read coverage of the contig was carried out using bwa-mem (http://bio-bwa.sourceforge.net), and then contig binning was done using MetaBAT-2 [29] with contigs longer than 2000 bp. The completeness and contamination level of the combined genomic bins were then evaluated using CheckM [30]. Only bins with a contamination level under 5% were analyzed. Genetic composition of genomic bins was then explored using KEGG [31] and MetaCyc [32] pathway mappers with genes identified by IMG/MER in the co-assembly (accession number: 3300038389). 

### 2.3. Quantitative PCR

The abundance of microbial 16S rRNA genes was estimated using real-time quantitative PCR (qPCR) with the same primer as used for amplicon sequencing (U515New_F and U926New). Additional qPCR experiments targeting the sulfate reducing microbial populations were performed using a combination of primers targeting the dsrB gene (DSR1728f-Mix/rDSR4r-Mix) as previously detailed [33]. Amplification reactions were performed in triplicate with a Rotor-Gene Q system (Qiagen, Hilden, Germany) in a final volume of 25 µL using Brilliant III SYBR SuperMix (Agilent Technologies, Santa Clara, CA, USA), 0.5 µM of each primer and 1 ng of DNA template. Standard curves from 10^6^ to 10^2^ copies of 16S rRNA and *dsrB* genes were prepared in triplicate with dilutions from 2.3 ng to 0.23 pg of genomic DNA from *Desulfobulbus propionicu*s (DSM2032) per reaction. The R^2^ values for standard curves obtained by real-time PCR were all greater than 0.997. qPCR efficiencies were above 92%.

## 3. Results

### 3.1. Geochemistry of Fluids from the Subsurface Gravity-Based Structure

Redox potential measured in the fluid of the gravity-based structure was −31.8 mV, indicating an anoxic and reducing environment. Temperature and conductivity of the fluid were similar to the ambient seawater temperature and conductivity in the North Sea with 10 °C and 45.5 mS/cm, respectively. The fluid was relatively poor in electron acceptors and donors with nitrate, nitrite, iron and manganese concentrations below detection limits (Figure 1 and Appendix A). Sulfate concentration was 1005 mg/L (approximately 40% of seawater sulfate ~2650 mg/L [34]). Hydrogen sulfide concentrations were below 5 ppm. Concentrations of volatile fatty acids, including acetic, propanoic, butyric, valeric and hexanoic acids were below the detection limit (0.1 mg/L). Total petroleum hydrocarbon concentration in the fluids from the gravity-based structure ranged 19.4–24.3 mg/L. Low molecular weight aromatic hydrocarbons such as benzene and toluene were present at up to 3.2 and 1.2 mg/L, respectively. Xylene, naphthalene and ethyl-benzene were also detected in the fluid but in lower concentrations (Figure 1 and Appendix A).

### 3.2. Microbial Community Composition 

Quantitative PCR targeting microbial 16S rRNA genes indicated an average of 1.6 × 10^6^ 16S rRNA gene per mL in the fluid from the GBS (Figure 2). Microbial community composition was determined using 16S rRNA gene amplicon and metagenomic sequencing with an average of 24,563 and 34,710 sequences per sample for amplicon and metagenome datasets, respectively. In total, 775 different operational taxonomic units (OTUs, 97% similarity) and 184 genera were identified by 16S rRNA genes sequencing (Appendix A). Although the relative proportion of the most predominant lineage (*Dethiosulfatibacter*) was different in the two samples analyzed and depending on whether metagenomes (31% of the 16S rRNA reads) or 16S rRNA genes amplicon sequencing (91% of the 16S rRNA gene amplicons) were analyzed, all analyses of 16S rRNA gene sequences showed a similar ranking of the most abundant lineages. Different members of the *Clostridiales* order (43–71% of the 16S rRNA reads in metagenomes), including *Dethiosulfatibacter* lineage (31–57% of the 16S rRNA reads) predominated the GBS communities (Figure 2 and Appendix A). Sequences affiliated with *Atribacteria* (2–5%), *Bacteroidetes* (0.4–5%), *Cloacimonadetes* (1.8–3.8%), *Spirochaeta* (2.5–6%), *Synergistales* (1–3%) and *Deltaproteobacteria* (4–9.5%), represented primarily by non-sulfate reducing *Pelobacter* species (1.9–3.5%), were also prominent members of the microbial communities in the GBS fluid (Figure 2). Archaea were mainly represented by various methanogenic lineages (1.2–3.2% based on 16S rRNA reads in metagenomes), including *Methanosaeta* genus (0.6–1.42%) (Figure 2). Only one read of 18S rRNA gene, affiliated to the marine worms *Distigma*, was recovered from the metagenomic dataset. If this sequence could also result from extracellular environmental DNA sequencing, this suggests that microbial eukaryotes were rare in the system.

### 3.3. Metabolic Pathways for Hydrocarbon Degradation

Two shotgun metagenomes from Cell6 of the GBS were sequenced to explore the metabolic potential of the microbial community. In total, 6572 different genes (KEGG Orthologues) were identified in the Cell6 metagenomes with various genes involved in hydrocarbon degradation. Comparison between numbers of aerobic and anaerobic alkane oxidation genes (*alkB* and *bssA* genes, respectively), identified in metagenomes, indicated that anaerobic hydrocarbon degradation pathway was predominant in the Cell6 fluid (4 *alkB* vs. 96 *bssA* genes). Nearly complete metabolic pathways for anaerobic degradation of toluene, ethylbenzene, phenol and naphthalene were identified in metagenomes (Figure 3). The metabolic pathway for the subsequent anaerobic degradation of benzoyl-CoA was also detected. Numerous alkyl succinate synthase genes (*assA/masD*) involved in anaerobic alkane degradation were also present in the metagenomes as well as acetylene hydratase genes involved in acetylene degradation (Figure 3).

### 3.4. Metagenome Assembled Genomes of GBS Populations

Due to the lack of close representative in databases, taxonomic information recovered from single genes was limited. We therefore used a genome-centric approach to identify the hydrocarbon degrading lineages in the system. In total, 27 medium to high quality metagenome assembled genomes (MAGs) (CheckM contamination < 3.3%, completeness > 43%) were recovered from the metagenomic data (Figure 4). Taxonomic affiliations of these MAGs, inferred from the 16S rRNA genes and ribosomal protein genes indicated that MAGs from predominant and rare lineages of the GBS fluids were reconstructed, including MAGs of *Clostridiales*, *Cloacimonadia*, *Deltaproteobacteria*, *Acetothermiia*, *Bacteroidetes*, *Synergistales* and *Archaea* lineages (Figure 4). Average coverage of the contigs ranged from 3.1× for one *Actinobacteria* bin to 490× for one *Clostridiales* bin likely related to *Dethiosulfatibacter*, detected as predominant member of the bacterial community by 16S rRNA genes analysis (Figure 2). Five MAGs were affiliated to Archaea. Metabolic pathways for methanogenesis from acetate and H_2_/CO_2_ were identified in MAGs related to *Methanosaeta*, *Methanolinea* and *Methanoculleus*, whereas genes for methanogenesis from methanol were detected in an unclassified Archaea taxonomically close to the methanogenic Archaea NM3 (Figure 4). Analysis of the bacterial MAGs revealed that fermentation and acetate metabolism were widespread in the bacterial community with numerous genes coding for the lactate dehydrogenase, formate dehydrogenase, alcohol dehydrogenase, acetate kinase and phosphate acetyltransferase identified in the MAGs, whereas other anaerobic energy producing pathways such as nitrate (*narG*, *napAB* and *nrfAH*) or sulfate (*dsrAB* genes) reductions were limited (Figure 4 and Appendix A). Potential end-products of these fermentation reactions include acetate, formate, lactate and ethanol, as well as CO_2_ and hydrogen with genes coding for hydrogenase complexes being identified in 90% of the MAGs (Figure 4). Furthermore, genes of the Rnf complex, a respiratory hydrogenase and the formate dehydrogenase involved in syntrophic metabolisms [35,36] were detected in numerous MAGs related to predominant lineages of the GBS fluid, including bins affiliated with *Syntrophus*, *Pelobacter* and *Cloacimonadia* for the Rnf complex and *Deltaproteobacteria*, *Synergistales* and methanogens MAGs for the formate dehydrogenase (Figure 4). Genes coding for multiheme cytochromes, potentially involved in direct electron transfer, were identified in *Pelobacter*, *Anaerolineae* and *Actinobacteria* MAGs. 

Genes for hydrocarbon degradation pathways, notably for aromatic hydrocarbon degradation genes (*HcrA1 HcrA2*, *HcrA3* and *HcrB* encoding for the 4-hydrocybenzoyl-CoA reductase subunits and BadDEG genes encoding subunit of the benzoyl-CoA reductase) were identified in 63% of the bacterial MAGs including in the predominant *Clostridiales/Dethiosulfatibacter* and *Cloacimonadia* bins. Alkyl-succinate synthase gene, involved in anaerobic alkane degradation, was found in *Pelobacter*, *Acetothermia* and *Synergistales* bins as well as in the *Clostridiales/Dethiosulfatibacter* bin (Figure 4). However, no complete pathways for anaerobic hydrocarbon degradation were recovered in a single genomic bin.

## 4. Discussion

Decommissioning of the Brent oil field Delta platform provided a unique opportunity for a detailed investigation of the microbial community structure and potential metabolism in these oil-impacted man-made subsea structures. Fluids from Cell6 of a concrete gravity-based structure were analyzed by qPCR, 16S rRNA genes sequencing and metagenomic sequencing coupled to geochemical characterization (Figure 2). Microbial abundance in the concrete structure was similar to sea water [37] with 1.6 × 10^6^ 16S rRNA genes per mL, corresponding to 1.33 × 10^6^ microbial cell per mL based on the average number of ribosomal operons found in MAGs (1.21), indicating a significant microbial biomass in the GBS. A diverse but uneven microbial community was detected, with anaerobic lineages previously found in oil reservoirs and marine seeps being present (Figure 2) [9,10,38,39]. This is consistent with the reductive, hydrocarbon-rich geochemical environment of the GBS. However, the microbial community was distinct from oil reservoir and marine seep communities, since lineages of the *Dethiosulfatibacter* and *Cloacimanadia* are usually found in very low proportion in these ecosystems [39]. Although this initial survey has examined a limited number of samples, the results suggest that hydrocarbon-degrading communities in man-made marine structures might diverge from those seen in natural hydrocarbon-rich environments, warranting further investigations of man-made ecosystems. 

### 4.1. The Guild of the Hydrocarbon Degraders

Genomic bin reconstruction revealed a large variety of microorganisms with genes associated with hydrocarbons degradation (Figure 4). Consistent with the dominant hydrocarbons present in the fluids, genes for low molecular weight aromatic hydrocarbon (benzene and toluene) degradation were identified in the metagenomic dataset and in numerous MAGs (Figure 1, Figure 2 and Figure 4). These results suggest that GBS fluid harbors a complex guild of hydrocarbon degraders composed of taxonomically different lineages including *Syntrophus* and *Desulfoglaeba* members which have cultivated representatives that are known alkane-degraders [40,41]. In addition, *Clostridiales* and *Anaerolineae* taxa, previously identified in hydrocarbon-degrading enrichments [11,42], and *Synergistales* and *Marinimicrobia,* previously suspected to degrade hydrocarbons based on metagenomic analyses [9], were also inferred to have the capacity for anaerobic hydrocarbon degradation. MAGs of the predominant *Claocimonadia* MSBL2 and *Dethiosulfatibacter* lineages also include genes from aromatic hydrocarbon degradation pathways as well as genes for alkane and ethylene degradation for *Dethiosulfatibacter.* These data indicate that the microbial community has potential for growth on a large number of hydrocarbons present in the system. Comparison of full length *Dethiosulfatibacter* 16S rRNA gene assembled from metagenomic reads against the NCBI database indicated that the most closely related sequences (FN550062, 98.76% of similarity over 1031 bp) were recovered from methane seep sediments [43], supporting the hydrocarbon degradation potential for these marine anaerobic taxa. 

### 4.2. Routes for Hydrocarbon Degradation and Bioremediation 

Hydrocarbon degradation in oil-impacted, anoxic, marine environments is frequently carried out by sulfate reducing microorganisms (SRMs) [44,45,46,47,48]. MAGs of SRMs recovered from the metagenomes (*Desulfoglaeba*, *Desulfomicrobiaceae* and *Desulfobacteriaceae*) included genes for hydrocarbon degradation, supporting these previous observations (Figure 4). However, data from 16S rRNA genes-based community composition analysis (Figure 2), coverage of the MAGs from SRMs (Figure 4) and numbers of *dsrB* genes in metagenomes indicate a lower proportion (~3%) of sulfate reducers in the community compared to other oil-impacted, anoxic marine ecosystems [33]. This result is also supported by previous *dsrB* gene quantification in these samples, which reported an average of 1.05 × 10^5^
*dsrB* genes in the system [33]. Reported to 16S rRNA gene number this suggests that less than 10% of the bacteria have the *dsrB* gene. These results, coupled with the absence of hydrogen sulfide, despite the significant amount of sulfate in the fluid, suggest that hydrocarbon degradation fueled by sulfate reduction was probably not the main hydrocarbon degradation route in the GBS fluid. By contrast, the proportion of potential hydrocarbon-degrading lineages with fermentative metabolism, such as *Clostridiales*, *Cloacimonadia*, *Pelobacter* and *Synergistales*, was greater than the proportion sulfate reducers in the system (69% vs. 3% of the community) (Figure 4). Sulfate reduction is energetically more favorable than fermentative metabolism [49], therefore the predominance of fermenters in GBS fluid was unexpected but could result from nutrient depletion or specific inhibition by potential biocide treatments used to control sulfide production at the oil platform.

Multiple lineages, including *Pelobacter*, *Syntrophus*, *Desulfoglaeba*, *Anaerolineae*, *Clostridiales* or *Sphaerocheta,* detected in the GBS have been associated with syntrophic lifestyles or enriched in hydrocarbon degrading co-cultures [4]. Likewise, *Acetothermia* and *Marinimicrobia* have been detected in synergistic networks involved in degradation of terephthalate in methanogenic bioreactors [50]. The genetic basis of syntrophic interactions and interspecies electron/metabolite exchange remains poorly characterized, but genome comparisons between syntrophic and non-syntrophic lineages have suggested a major role of the formate dehydrogenase and hydrogenases, notably the Rnf complex, in syntrophy and interspecies energy transfer [12,35,51,52]. Genes coding for the formate dehydrogenase as well as for FeFe hydrogenases, hydrogenases of Types 1, 3 and 4 and the Rnf complex were identified in most of the GBS MAGs, supporting syntrophic capabilities. It also suggests that any degraded hydrocarbons might be fermented to volatile fatty acids (formate and acetate) and hydrogen. These products, in turn, might be consumed by hydrogenotrophic, formate- and acetate-utilizing members of the community after interspecies exchanges. 

Although 16S rRNA gene analyses of the metagenomic data indicated that archaea represented a low relative proportion of the microbial community (1.2–3.2%), the aceticlastic and hydrogenotrophic methanogenic capabilities detected in the archaeal MAGs suggest that these archaea could play a role in the hydrocarbon degradation processes through hydrogen and acetate scavenging, as previously observed in other anaerobic alkane-rich systems [12,52]. Detection of formate dehydrogenase in *Methanomicrobiales* bins also indicated the potential for formate utilization and interspecies formate transfer as syntrophic mechanism, as previously reported [52,53]. Interestingly, one unclassified archaeal bin included a divergent mcrA gene as well as genes for methanol-based methanogenesis. Although it remains unclear whether this divergent mcrA gene is involved in methanogenesis or short alkane degradation [54], sequences of this bin showed similarity with sequences of the *mcrA* containing-Archaea NM3 bin, supporting the presence of a divergent *mcrA* gene in this new group of uncultivated archaea [54]. NiFe hydrogenase genes were also identified in our archaeal bin, suggesting a potential role in hydrogen turnover and scavenging for this uncultivated archaeon. 

In addition, genes for multiheme c-type cytochromes, potentially involved in direct interspecies electron transfer as well as in metal oxides reductions [55], were identified in MAGs related to *Pelobacter, Anaerolineae* and *Actinobacteria*. Genes for hydrocarbon degradation were also identified in *Pelobacter* and *Anaerolineae* MAGs, suggesting that syntrophic hydrocarbon degradation processes might also involve direct interspecies electron transfer between members of the microbial community.

### 4.3. Implication for Decommissioning Process of Gravity-Based Substructures

Although more samples are required to achieve a complete view of the concrete gravity-based structures microbiome, our genomic investigation of the GBS fluid provided an unprecedented overview of the microbial processes that may occur within oil-containing GBS and such insight could potentially influence future decommissioning projects of GBSs. Conventional assessments of microbial community composition, abundance and activity in the oil industry are often limited to culture-based approaches such as MPN counts of broad groups of microorganisms and quantification of metabolic end-products (e.g., H_2_S and volatile fatty acids). The very low levels of H_2_S (below the 5 ppm that can be detected with lead-acetate paper) and volatile fatty acids (<0.1 mg/L) coupled with the very low number of sulfate reducing bacteria obtained by MPN quantification present in the GBS has been interpreted as indicating that hydrocarbon degradation is unlikely. However, our genomic results indicate a strong hydrocarbon degradation potential in the GBS fluid. This potential was not associated with sulfate reduction but with fermentations and syntrophic metabolism. Therefore, the low level of volatile fatty acids might indicate a high turnover of these compounds during syntrophic microbial hydrocarbon degradation rather than an inactive hydrocarbon-degrading community. Although these results must be confirmed by activity-based approaches, our results expand the list of potential hydrocarbon degraders and reveal that hydrocarbon-degrading communities in man-made marine structures might differ markedly from oil-impacted anoxic natural environments. Our study therefore illustrates the importance of considering genomic data in any future research into the feasibility of bioremediation of GBS fluids.

## Figures and Tables

**Figure 1 microorganisms-09-00356-f001:**
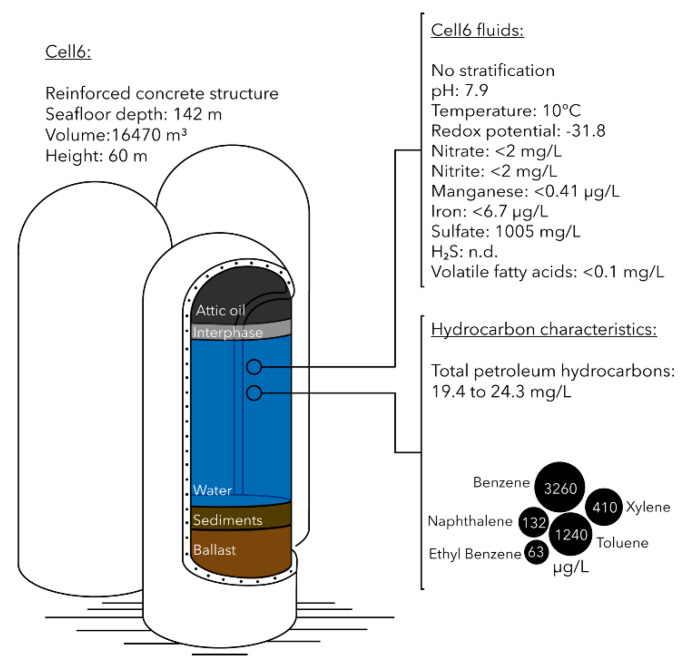
Schematic description of Cell6 from the gravity-based structure of Brent oil and gas platform, with fluid characteristics and hydrocarbon composition.

**Figure 2 microorganisms-09-00356-f002:**
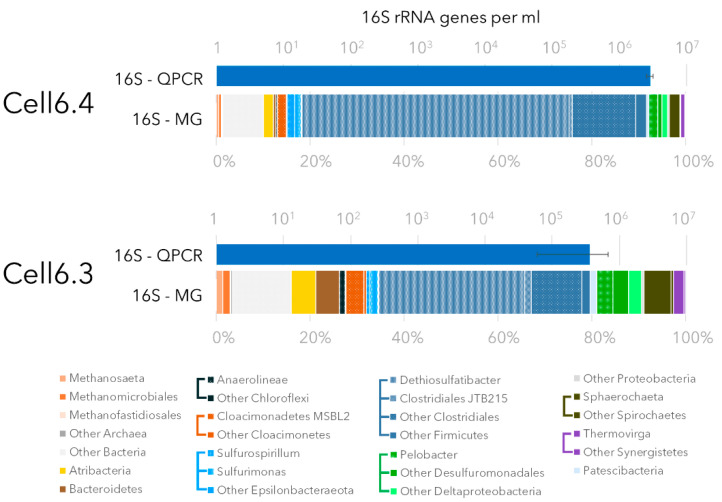
16S rRNA gene abundance and community composition in the GBS (Cell6.4 and Cell6.3 samples). Number of 16S rRNA gene was determined by quantitative PCR. Scale is logarithmic. Microbial community composition is based on 16S rRNA genes recovered from the metagenomic dataset. Lineages are color-coded by phyla with *Chloroflexi* in black, *Cloacimonetes* in Orange, *Epsilon*/*Campylobacteria* in light Blue, *Firmicutes* in dark Blue, *Deltaproteobacteria* in Green, *Spirochaetes* in brown and *Synergistetes* in purple.

**Figure 3 microorganisms-09-00356-f003:**
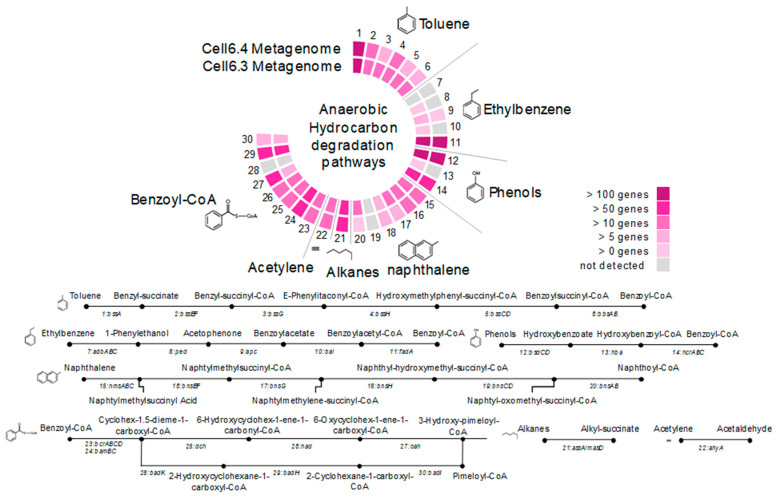
Metabolic pathways for anaerobic hydrocarbon degradation identified in the metagenomic dataset: (1) *bssA*, benzylsuccinate synthase (K07540); (2) *bssEF*, benzylsuccinate CoA-transferase (K07543); (3) *bssG*, benzylsuccinyl-CoA dehydrogenase (K07545); (4) *bssH*, phenylitaconyl-CoA hydratase (K07546); (5) *bssCD*, hydroxy(phenyl)methyl-succinyl-CoA dehydrogenase (K07547); (6) *bbsAB*, benzoylsuccinyl-CoA thiolase (K07549); (7) *abdABC*, ethylbenzene hydroxylase (K10700); (8) *ped*, phenylethanol dehydrogenase (K14746); (9) *apc*, acetophenone carboxylase (K10701); (10) *bal*, benzoylacetate-CoA ligase (K14747); (11) *fadA*, acetyl-CoA acyltransferase (K00632); (12) *bsdCD*, vanillate/4-hydroxybenzoate decarboxylase (K01612); (13) *hba*, hydroxybenzoate-CoA ligase (K04105); (14) *hnABC*/*HcrABC*, hydroxybenzoyl-CoA reductase (K04107); (15) *nmsABC*, naphthyl-2-methylsuccinate synthase (K01670); (16) *bnsEF*, naphthyl-2-methylsuccinate CoA transferase (K15569); (17) *bnsG*, naphthyl-2-methylsuccinyl-CoA dehydrogenase (K15571); (18) *bnsH*, naphthyl-2-hydroxymethylsuccinyl-CoA hydratase (K15572); (19) *bnsCD*, naphthyl-2-hydroxymethylsuccinyl-CoA dehydrogenase (K15573); (20) *bnsAB*, naphthyl-2-oxomethyl-succinyl-CoA thiolase (K15574); (21) *assA*/*masD*, alkylsuccinate synthase; (22) *ahy*, acetylene hydratase (K20625); (23) *bcrABCD*, benzoyl-CoA reductase (K04112); (24) *bamBC*, benzoyl-CoA reductase (K19515); (25) *dch*, cyclohexa-1,5-dienecarbonyl-CoA hydratase (K07537); (26) *had*, 6-hydroxycyclohex-1-ene-1-carbonyl-CoA dehydrogenase (K07538); (27) *oah*, 6-oxocyclohex-1-ene-carbonyl-CoA hydrolase (K07539); (28) *badK*, cyclohex-1-ene-1-carboxyl-CoA hydratase (K07534); (29) *badH*, 2-hydroxycyclohexanecarboxyl-CoA dehydrogenase (K07535); and (30) *badI*, 2-ketocyclohexanecarboxyl-CoA hydrolase (K07536).

**Figure 4 microorganisms-09-00356-f004:**
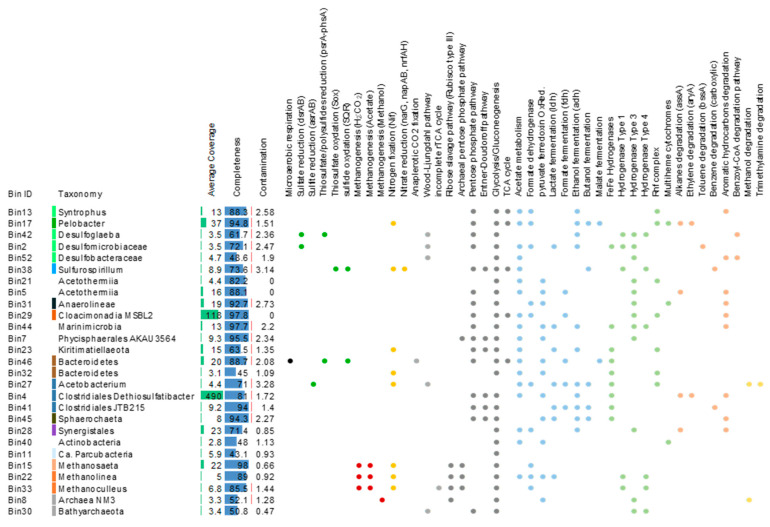
Description of the MAGs recovered from the metagenomes. Taxonomy of the MAGs was determined using ribosomal proteins and completeness and contamination were estimated using CheckM. Average coverage corresponds to the average of the coverage of all contigs included per MAG. MAGs are color coded (at the left of the taxonomic names) per classes using the same color code as in Figure 1. Raw data are available in Appendix A.

## Data Availability

Raw 16S rRNA gene sequences are available on NCBI website using Bioproject ID: PRJNA472634. Metagenomes are available in IMG/M under the following accession numbers: 3300010410 and 3300010411. Combined metagenome is also available in IMG/M using 3300038389 accession number.

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
