# Peer review of "Syntrophic Hydrocarbon Degradation in a Decommissioned Off-Shore Subsea Oil Storage Structure"

_microorganisms, 2021, doi:10.3390/microorganisms9020356_

Round 1

Reviewer 1 Report

The quality of the paper was significantly improved, nice work.

But there are two major points that I consider relevant to proceed to the publication of the paper:

  • In the last revision:

“Figure 2 represents the results from 16S rRNA gene sequencing showing both the taxonomic

identification and the relative abundance. However, one should be able to have access to the complete data, the full identification and quantification (i.e., the identification of each taxon with the respective percentages). I suggest that the authors organize the results in a big table with this information and provide as supplementary results (it can be an excel file).”

Authors response:

“As requested, an excel file with the number of sequences after normalization and taxonomic affiliation for each OTU has been added in the supplementary material. In addition, raw sequence data are available on NCBI website.”

However, the excel file was not provided with the revised version and this is an important issue.

  • In the discussion there is the reference to a number of genes that were found and assigned to certain MAGs. This description however is not supported by the results presented. It is crucial to present those results. Again I suggest the presentation of an excel table per MAG with the list of the genes identified. This is not much work since the authors probably have already the data organized as it was necessary to write the results and discussion sections. Figures 3 and 4 are related to this issue but do not replace the complete list of genes identified in each MAG.

Minor changes/comments:

Line 72: Gammaproteobacteria in italic.

Line 94: Pili and nanowires are not the same thing, please check the meaning and change the text accordingly.

Figure 3: congratulations for the changes in Figure 3.

Author Response

Reviewer # 1

The quality of the paper was significantly improved, nice work.

But there are two major points that I consider relevant to proceed to the publication of the paper:

In the last revision:

“Figure 2 represents the results from 16S rRNA gene sequencing showing both the taxonomic identification and the relative abundance. However, one should be able to have access to the complete data, the full identification and quantification (i.e., the identification of each taxon with the respective percentages). I suggest that the authors organize the results in a big table with this information and provide as supplementary results (it can be an excel file).”

Authors response:

“As requested, an excel file with the number of sequences after normalization and taxonomic affiliation for each OTU has been added in the supplementary material. In addition, raw sequence data are available on NCBI website.”

However, the excel file was not provided with the revised version and this is an important issue.

The excel file was provided in the compressed supplementary material as Supplementary Table S2.xlsx (now S3). This is the journal policy to upload all the supplementary material in a single archive file and we don’t have the possibility to upload all files separately.

In the discussion there is the reference to a number of genes that were found and assigned to certain MAGs. This description however is not supported by the results presented. It is crucial to present those results. Again I suggest the presentation of an excel table per MAG with the list of the genes identified. This is not much work since the authors probably have already the data organized as it was necessary to write the results and discussion sections. Figures 3 and 4 are related to this issue but do not replace the complete list of genes identified in each MAG.

We understand the reviewer concern. To answer this point, we have now uploaded an excel file with the list (and number) of Kegg orthologies (KO) found in each MAG in the supplementary material (Supplementary Table S4.xlsx).

Minor changes/comments:

Line 72: Gammaproteobacteria in italic.

Since we are using the adjective “Gammaproteobacterial”, we believe that it should not be in italic.

Line 94: Pili and nanowires are not the same thing, please check the meaning and change the text accordingly.

We agree with the reviewer that nanowires are not pili but stacked cytochromes in a pili-like structure. The sentence has been modified. (Line 72)

Figure 3: congratulations for the changes in Figure 3.

Thank you very much for your positive feedback

Reviewer 2 Report

Second round of revision

I'd like to thank the authors for their thoughtful consideration of most of my review comments and appreciate their further efforts, which I feel have strengthened the manuscript.

At the same time, I would like to note that the Authors decided not to follow some of my suggestions, and this in my opinion significantly reduces the final scientific value of their work. However, this was their choice and I think this does not preclude the possibility of publication.

Just a few things that need to be fixed or considered:

  1. The "W" modification in the primer must be supported by appropriate evidence (data from the in silico comparison of the different primers). This modification cannot be accepted as a personal communicaton by the Authors. AND, this aspect has to be checked also using the Silva 138.1 relaese (info must be provided as supplementary).
  2. The reply to my comment about the metagenomics vs 16S metabarcoding approach cannot be accepted.. I suggested to join the two figures and maybe discuss the differences if using the different approaches (which could highlight some interesting aspects related to the primer coverages in these sepcific samples, if correctly analyzed). Defending one approach against the other is not correct here, as both have significant flaws. However, no strict need to put more efforts on this aspect if the Authors think it is not worth it. At the same time, the color legend of Supp Fig 1 must be, at least, made consistent with that in the maintext (Fig 2), and I am not convinced that the revised color legend is sufficiently clear.. please clearly write/show the correspondence between the highest taxonomy level used (phylum?) and its related color.
  3. I noticed confusion between microbial cell abundance and abundance of 16S genes. The authors should consider the number of 16S gene copies per cell and convert gene abundance into estimates of cell abundance, based on what curently known on this topic(see https://rrndb.umms.med.umich.edu/)
  4. L240 Standard curves from 106 to 102 copies... corresponding to what quantity of genomic DNA?
  5. I might have missed it, but I think that the Authors did not reply to one of my questions (L157 How much DNA was extracted from the 60 ml filtered material?)
  6. In the Abstract, the last sentence could lead the reader to expect direct (novel) bioinformatic analyses to compare the microbial assemblage composition of different types of samples, which is not the case of this work. Please rephrase to avoid misunderstanding.
  7. L281 "Only one read of 18S rRNA gene [...] suggesting that microbial eukaryotes were rare in the system". The Authors could consider that this might also be extracellular DNA (see for example Corinaldesi et al 2018 doi.org/10.1038/s41598-018-20302-7), and possibly include some considerations about metazoa in such kind of anoxic systems (for example, see Danovaro et al., 2010 (doi.org/10.1186/1741-7007-8-30)

Author Response

I'd like to thank the authors for their thoughtful consideration of most of my review comments and appreciate their further efforts, which I feel have strengthened the manuscript.

At the same time, I would like to note that the Authors decided not to follow some of my suggestions, and this in my opinion significantly reduces the final scientific value of their work. However, this was their choice and I think this does not preclude the possibility of publication.

Just a few things that need to be fixed or considered:

  1. The "W" modification in the primer must be supported by appropriate evidence (data from the in silico comparison of the different primers). This modification cannot be accepted as a personal communicaton by the Authors. AND, this aspect has to be checked also using the Silva 138.1 relaese (info must be provided as supplementary).

The in silico analysis of the primer (including against Silva v138) has been added in the supplementary material archive.

  1. The reply to my comment about the metagenomics vs 16S metabarcoding approach cannot be accepted.. I suggested to join the two figures and maybe discuss the differences if using the different approaches (which could highlight some interesting aspects related to the primer coverages in these sepcific samples, if correctly analyzed). Defending one approach against the other is not correct here, as both have significant flaws. However, no strict need to put more efforts on this aspect if the Authors think it is not worth it. At the same time, the color legend of Supp Fig 1 must be, at least, made consistent with that in the maintext (Fig 2), and I am not convinced that the revised color legend is sufficiently clear.. please clearly write/show the correspondence between the highest taxonomy level used (phylum?) and its related color.

The legend of the figure has been modified to detail the color code and caption of the supplementary figure has been modified to be consistent with the main figure.

  1. I noticed confusion between microbial cell abundance and abundance of 16S genes. The authors should consider the number of 16S gene copies per cell and convert gene abundance into estimates of cell abundance, based on what curently known on this topic(see https://rrndb.umms.med.umich.edu/)

We agree with the reviewer that 16S rRNA gene numbers are only a proxy for the abundance and the number of copy per genome might be used for a better estimation of the abundance. However, the number of copy per genome is extremely variable (1 to 21) depending on the species. A quick search on the rrndb website resulted in no matches for all the dominant lineages of our system, therefore we analyzed the number of rRNA operon in our MAGs and found an average of 1.21 16S rRNA gene copies. After correction, the microbial abundance is now 1.33 x106 per mL. The manuscript has been modified to avoid confusion. (Caption of figure 2 and line 333)

  1. L240 Standard curves from 106 to 102 copies... corresponding to what quantity of genomic DNA?

106 to 102 copies correspond to 2.3 ng to 0.23 pg of genomic DNA per reaction. This has been added in the text line 203

  1. I might have missed it, but I think that the Authors did not reply to one of my questions (L157 How much DNA was extracted from the 60 ml filtered material?)

Based on Qubit measurements an average of 2.19 ng of DNA per µL was extracted from the filters. This has been added in the text line 138.

  1. In the Abstract, the last sentence could lead the reader to expect direct (novel) bioinformatic analyses to compare the microbial assemblage composition of different types of samples, which is not the case of this work. Please rephrase to avoid misunderstanding.

The sentence has been modified to avoid misunderstanding (Line 40)

  1. L281 "Only one read of 18S rRNA gene [...] suggesting that microbial eukaryotes were rare in the system". The Authors could consider that this might also be extracellular DNA (see for example Corinaldesi et al 2018 doi.org/10.1038/s41598-018-20302-7), and possibly include some considerations about metazoa in such kind of anoxic systems (for example, see Danovaro et al., 2010 (doi.org/10.1186/1741-7007-8-30)

We agree with the reviewer, it is likely that this sequence result form environmental DNA sequencing. This has been added in the text line 245.